# Associations of 16-Year Population Dynamics in Range-Expanding Moths with Temperature and Years since Establishment

**DOI:** 10.3390/insects14010055

**Published:** 2023-01-06

**Authors:** Per-Eric Betzholtz, Anders Forsman, Markus Franzén

**Affiliations:** Department of Biology and Environmental Science, Linnaeus University, SE-39182 Kalmar, Sweden

**Keywords:** abundance, climate change, light trap, migration, moths, population growth, Sweden, temperature

## Abstract

**Simple Summary:**

There has been a widespread decline of many plants and animals driven at least partly by climate change. This pattern is not universal, and certain taxa are increasing in abundance and distribution. A better understanding of population dynamics and range expansions in different areas and how different taxa respond to changing temperatures is therefore important, as we are facing a warmer and more fluctuating climate in the future. In this study, we show that range-expanding moths in southeastern Sweden have increased their species richness over time and that abundance and population growth increase during years with higher temperatures. We also show that population growth in range-expanding moths is fastest in the first years after establishment in an area. These shifts in distribution and abundance of moths may lead to rapid and dramatic changes in community compositions, with potentially widespread consequences for species interactions and ecosystem functioning.

**Abstract:**

Parallel to the widespread decline of plants and animals, there is also an ongoing expansion of many species, which is especially pronounced in certain taxonomic groups and in northern latitudes. In order to inform an improved understanding of population dynamics in range-expanding taxa, we studied species richness, abundance and population growth in a sample of 25,138 individuals representing 107 range-expanding moth species at three light-trap sites in southeastern Sweden over 16 years (from 2005 to 2020) in relation to temperature and years since colonisation. Species richness and average abundance across range-expanding moths increased significantly over time, indicating a continuous influx of species expanding their ranges northward. Furthermore, average abundance and population growth increased significantly with increasing average ambient air temperature during the recording year, and average abundance also increased significantly with increasing temperature during the previous year. In general, population growth increased between years (growth rate > 1), although the population growth rate decreased significantly in association with years since colonisation. These findings highlight that, in contrast to several other studies in different parts of the world, species richness and abundance have increased in southeastern Sweden, partly because the warming climate enables range-expanding moths to realise their capacity for rapid distribution shifts and population growth. This may lead to fast and dramatic changes in community composition, with consequences for species interactions and the functioning of ecosystems. These findings are also of applied relevance for agriculture and forestry in that they can help to forecast the impacts of future invasive pest species.

## 1. Introduction

A widespread decline in species richness, abundance and biomass of insect species has recently been reported in several studies [1,2,3,4,5,6,7], affecting insect–plant interactions [8] and ecosystem functioning [9]. However, some studies indicate that this pattern is not universal; some taxa and sites show decreases in abundance and diversity, whereas others have increased or remained unchanged [10,11,12,13,14,15]. In northern latitudes, there is also an ongoing range expansion of moths that is likely driven, at least partly, by a warming climate [16,17,18]. A better understanding of population dynamics and range expansions in different areas and taxa and how they respond to changing temperatures is therefore important, specifically given that we are facing an even warmer and more fluctuating climate in the future [19,20,21,22].

Large moths are well-suited for exploring dynamics in range-expanding species. Moths constitute a large part of terrestrial biodiversity in terms of species richness and form the functional link between producers and predators in food webs. They also display a wide variety in their feeding spectrum and respond quickly to climatic and environmental changes; furthermore, there is a solid and robust knowledge of their ecology, especially in northern Europe. Range expansions may result in the rapid emergence of novel and drastically altered species communities [15]. Some range-expanding moth species are invasive, may exhibit extreme population growth and potentially become serious pests in their new habitats [23,24,25,26].

Despite the impact that moths may have on ecosystem functioning, relatively few studies have investigated the drivers of population dynamics in range-expanding species of moths [11]. To inform a better understanding of these issues, we explored how between-year variation in species richness, abundance and population growth are associated with temperature and how population growth changes in association with time since colonisation. To address these questions, we analysed light-trap data of moths from three sites in southeastern Sweden, collected over 16 years (from 2005 to 2020) and consisting of >25,000 individuals representing 107 range-expanding species.

## 2. Materials and Methods

### 2.1. Species, Locations and Climate

We used a dataset from three light-trap sites situated in southeastern Sweden, i.e., Nedra Ålebäck and Össby in the province of Öland and Utlängan in the province of Blekinge (Figure 1a). The dataset covers 16 years from 2005 to 2020, with the aim of surveying range-expanding species, including migrating, partly migrating and new species to the region (Figure 1b). The location of the light traps in southeastern Sweden is very suitable for monitoring expanding species originating from the European continent or the very southern parts of Sweden, i.e., the province Scania. Furthermore, light-traps were deployed along field margins, offering excellent opportunities to survey expanding species and their dynamics [27]. The Ryrholm-type light traps [28] were operated with 125 W Philips mercury vapour lamps each night from May to October between 2005 and 2020. The traps were emptied every second to third week throughout the study period. To obtain a high-quality dataset, all individuals were counted and identified by species by one of the authors (P.E.B.), an experienced moth identifier. Taxonomy and systematic criteria were implemented according Aarvik et al. [29].

Utlängan (56.022731 N/15.797629 E) is an island with an area of 215 ha situated 7 km southeast of the mainland (Figure 1). Habitats on the island are dominated by woods and meadows. The trap at Utlängan is situated 3 m above sea level 210 m from the coastline of the Baltic Sea, and seminatural grasslands occur between the trap and the sea. Össby (56.270783 N/16.490312 E) and Nedra Ålebäck (56.605853 N/16.686114 E) are small villages situated on the east coast of the island of Öland (Figure 1). Meadows and farmlands dominate the surroundings of both villages. The trap in Össby is situated 6 m above sea level 370 m from the Baltic Sea, and the trap in Nedra Ålebäck is situated 2 m above sea level and 820 m from the Baltic Sea. The distance to the reference WMO (World Meteorological Organization) meteorological station Ölands Södra udde is 48.9 km from Nedra Ålebäck, 10.1 km from Össby and 43.1 km from Utlängan. The distance between the most northern light trap position, Nedra Ålebäck, and the most southern trap at Utlängan is 92.0 km. The WMO station Ölands södra udde is situated at sea level 225 m from the coastline.

The general climate in the study region is characterised by rather cold winters and warm, dry summers. According to SMHI [30], the daily mean temperature is −1 to 2 °C during mid-winter and 16 to 17 °C during summer. The area is one of the driest in Sweden, with yearly mean precipitation of 450 mm. We extracted temperature data (2004–2020) from the WMO meteorological station at Ölands södra udde [31] (Appendix A), the southernmost point of the island of Öland (Figure 1). We used the average ambient air temperature from May to July according to Pöyry et al. [32]. Data from the meteorological station Ölands södra udde are suitable because they not reflects the weather situation of the study area, being close to the coastline and in the centre of the area where the light-traps are located.

### 2.2. Data Analysis and Statistics

First, we employed a general linear mixed model (GLMM) with a Poisson error distribution and log-link function to explore whether species richness (the number of range-expanding species per recording year and site) was associated with ambient air temperature. Species richness was the response variable, and the site was included as a random effect. We included the continuous variables year and temperature during the recording year (t) and the previous year (t − 1) as explanatory variables. The correlation between temperature, year (t) and (t − 1) was not significant (Pearson correlation = 0.301, *p* = 0.257, *n* = 16).

Second, we employed a GLMM with a Poisson error distribution and log-link function to explore whether abundance (the number of individuals per species, year and site) was associated with ambient air temperature and year. Abundance was the response variable, and site and species were included as random effects. We included the continuous variables year and temperature during the recording year (t) and the previous year (t − 1) as explanatory variables.

Third, we employed a GLMM to explore whether population growth depended on the number of years after colonisation of a trap site and whether years since establishment was associated with ambient air temperature. Population growth (natural log-transformed) was the response variable, and site and species were included as random effects. As explanatory variables, we included the continuous variables years after colonisation, calculated as the number of years between the recording year and the year the species was first recorded at a trap site, and temperature during the recording year (t) and the previous year (t − 1). With this analytical approach, population growth (natural log-transformed) indicates no growth if the value is one, decreasing population growth if the value is <1 and increasing population growth if the value is >1. We also included the quadratic term of year since colonisation as an explanatory variable in the model to evaluate potential curvilinear relationships of population growth with time since colonisation. The glmmTMB package was used for the GLMMs [33]. Data were analysed using R [34].

## 3. Results

We recorded 25,138 individuals representing 107 range-expanding moth species at the three light-trap sites over 16 years between 2005 and 2020 (Appendix A). Most species were found in Nedre Ålebäck, with 88 species; followed by Utlängan, with 87 species; and Össby, with 79 species. All three sites were occupied by 58% (62) of species, and 22 species only occurred at one site. Out of the 107 species, 32 were recorded for the first time in the region in at least one of the provinces where the light-trap sites were situated. Across years, species richness in the traps varied between 13 and 56 species (mean = 32), whereas total abundance (per year and site) varied between 78 and 3047 individuals (mean = 524) (Figure 1c,d).

The three most abundant species during the study period were *Hoplodrina ambigua*, with 8083 individuals in total, ranging from 1 individual in 2006 to 3284 individuals in 2019; followed by *Noctua interjecta*, with 3268 individuals in total, ranging from 1 individuals in 2006 to 543 individuals in 2018; and *Phlogophora meticulosa*, with 1640 individuals in total, ranging from three individuals in 2010 to 366 individuals in 2015 (Appendix A).

Species richness and average abundance increased significantly over time, and both species richness (Table 1, Figure 1c) and average abundance (Table 2, Figure 1d) increased significantly with increasing average ambient air temperatures during the recording year. The average abundance also increased significantly with increasing temperature during the previous recording year (Table 2, Figure 2b).

Population growth increased significantly with increasing average ambient air temperatures during the recording year (Table 3, Figure 3a). Population growth was also positive across all years, indicating an increase in abundance from one year to the another, although the rate of population growth decreased significantly in association with years since the colonisation of a trap site (Table 3, Figure 3b).

## 4. Discussion

We found that both species richness and the average and total abundance of range-expanding moths in southeastern Sweden increased over a 16-year period (2005–2020) and that temporal changes in species richness, abundance and population growth were driven, at least partly, by temperature. These findings contrast the results and conclusions reported in several previous studies indicating a severe and ongoing decline in insect richness and abundance [2,3,7] and add to the number of studies indicating an increase in species richness and abundance in certain taxonomic groups and regions [11,12,14]. The increase in species richness and abundance over time reported in this study (Figure 1c,d) indicates a continuous influx of species to southeastern Sweden, expanding their ranges northwards. In agreement with our study, many moth species have a continuous northward range expansion and have colonised other countries in Europe such as Britain, Holland, Denmark and Finland, thus expanding their European range in response to climate change [35,36,37,38].

The results identify years with high ambient air temperature as an important driver increasing abundance and population growth, in agreement with Bowler et al. [39], who identified temperature as the most important predictor of recent population trends in terrestrial organisms. Although we did not find a statistically significant pattern of temperature change at the reference WMO station, Ölands södra udde, during the course of our study period from 2005 to 2020 (Appendix A; linear regression, b = 0.030 ± 0.0050, *t* = 0.505, *p* = 0.555), we observed a significant temperature increase of 1.5 °C in southeastern Sweden during the period of 1991–2020 compared to 1961–1990 [40], creating climatically suitable conditions that allow for thermophilic species to expand and establish in new areas [10,16,17,41]. At latitudes in our study area, climatically sensitive range-expanding species thrive during warm years, likely owing to increased adult survival, shorter larval development times and increased larval metabolism and survival [42,43]. Conversely, during years with lower temperatures further from the species’ thermal and physiological tolerance limits, range-expanding moths are more likely to suffer from declines in survival and population growth. An additional explanation for range expansions may be that vascular plants respond more slowly to climatic warming by shifting their ranges. Therefore, areas where host plants were already present but that used to be climatically unsuitable for moths have become suitable to moths due to warming.

During the study period, we recorded 32 species new to at least one of the provinces where the light traps were located, indicating an ongoing influx of species to the region. One of the species was actually the first record of the Swedish fauna *Eublemma purpurina*. Betzholtz and Franzén [44] previously showed that the recruitment areas from where expanding species originate and colonise into the studied region probably are continental Europe or the very southernmost province of Sweden, Scania, and that moths actually cover distances of the magnitude needed for those expansion and migration events. Furthermore, high intraspecific variation and generalist strategies typically promote establishment success and range expansions [11,45,46], but specialist species, e.g., nettle-feeding species, may be successful under certain conditions [16]. We recorded several species of expanding moths that were previously migratory in the region but that, during the study period between 2005 and 2020, established and increased in population. This apparent expansion, mainly driven by the factors discussed in the paragraph above, includes species from several taxonomic families, e.g., the pyralids *Sciota fumella*, *Nephopterix angustella* and *Pyralis regalis*; the geometrids *Idaea ochrata*, *Eupithecia pulchellata* and *Thedidia smaragadaria*; and the noctuids *Proxenus lepigone*, *Coenobia rufa*, *Athetia centrago* and *Agrotis puta* (cf. Appendix A). Some of these recently established species are already thriving in the region, e.g., the drepanid *Watsonalla binaria* and the noctuids *Cryphae algae*, *Haplodrina ambigua*, *Mythimna albipuncta*, *Noctua interjecta* and *Noctua interposita* (Appendix A). There are also several examples of species that we argue are currently establishing and expected to increase in population over the next decade, e.g., *Loxostege turbidalis*, *Udea accolalis*, *Diasemia reticularis* and *Lacanobia splendens*. There are also migrating species that irregularly but often reproduce in the region during summers with higher temperatures, e.g., the sphingiid *Hyles galii* and the noctuids *Cucullia fraudatrix*, *Cucullia artemisiae*, *Macdonnoughia confusa* and *Autographa mandarina*, indicating that these species as possible establishers in the future, given a continued increase in average ambient temperatures. We also recorded some species that had gone temporarily extinct from the Swedish fauna but recolonised the region during the study period. One of these species is *Cucullia scrophulariae*, which last had reproducing populations in the region back in the 1980s but is now reproducing again in the region. Another example is *Xestia ditrapezium*, a species that was recorded in the province of Scania in the early 1800s and now probably reproduces in small numbers in the southeastern parts of the province of Öland [47].

We also recorded several pest species that can potentially establish and become serious pests in the studied region due to the warming climate. The most prominent of these species is probably the erebid *Euproctis chryssorhoa*, a species that is a major pest in orchards of continental Europe. Periodically, i.e., from the 1950s to 1970s, the species has had temporary populations as far north as Denmark and the island of Ven, situated in Öresund between Denmark and Sweden [48]. We recorded a major influx of this species to the region in 2006. During the latter part of the study period, there were almost yearly records of this species in the light traps, indicating that we probably face an establishment of this species in the near future. Furthermore, we recorded seven noctuid species that are considered pests in their common habitat [26]: *Phlogophora meticulosa*, *Trichoplusia ni*, *Heliothis armigera*, *Spodoptera exigua*, *Agrotis bigramma*, *Agrotis ipsilon* and *Peridroma saucia*. Of these, *H. armigera* is a potent pest on a wide range of important cultivated crops and is expanding its northern range in Europe [49]. We recorded *H. armigera* almost yearly during the latter period of the study. This species has the potential to exhibit successful summer generations in Sweden, qualifying it as a potential threat to cultivation in our region. On the other hand, *T. ni*, *S. exigua*, *A. bigramma* and *P. saucia* should be considered true migrants and are not expected to be established in the region soon. The noctuids *P. meticulosa* and *A. ipsilon* have been established in the region for a long time, but they have not behaved as pests to the best of our knowledge. However, during warmer years, both species occur in higher numbers (Appendix A, Appendix A), indicating that they could also be of concern in terms of future damage to, e.g., vegetables and grains in our region. We also recorded four pyralid pest species, all of which were favoured during warm years, when they occurred in higher abundance and constituted a greater threat to agriculture. *Loxostege sticticalis* is a polyphagous pest that inflicts most harm on sugar beet and beans, *Udea ferrugalis* and *Nomophila noctuella* are pests on various crops, e.g., alfalfa fields. Furthermore, we recorded *Palpita vitrealis*, an important pest of olives in several Mediterranean countries, e.g., Italy [50]. However, the latter species should be of no concern as a pest, as olives are not grown in our region.

We found population growth to be positive throughout the study period and that the growth rate decreased significantly with time since the colonisation of a trap site. A pattern such as this may arise in response to parasitic interactions, referred to as the enemy release hypothesis [51,52]. This hypothesis predicts that in a newly colonised area, species experience a period during which they escape their natural enemies until interactions with the local parasitic complex are established [53,54] and the dynamics are stabilised. An additional explanation for population size being stabilised over time is resource limitation [11,18] of larval host plants and adult nectar plants. An example species that shows a growth pattern in agreement with this hypothesis is the noctuid *Eucarta virgo*, which peaked a few years after establishment in our region and that has now stabilised at lower population numbers (Appendix A).

It can be argued that a limitation of the present study is that we focused on a subset of range-expanding moths. Nevertheless, during the study period, we found a statistically significant and consistent increase in both species richness and the abundance of more than 100 range-expanding species in our dataset and not a no-change scenario.

## 5. Conclusions

This study highlights that the populations of range-expanding moths in southeastern Sweden have increased both in abundance and species richness over 16 years and that their potential for rapid population growth seems to be favoured by a warming climate. These shifts in distribution and average abundance of moths may lead to fast and widespread changes in community compositions. This may, in turn, have widespread consequences for species interactions and ecosystem functioning, given that moths are important pollinators of plants and food sources for birds and bats. These findings also have applied relevance for agriculture and forestry when forecasting impacts from invasive species potentially becoming serious pests within their new habitats.

## Figures and Tables

**Figure 1 insects-14-00055-f001:**
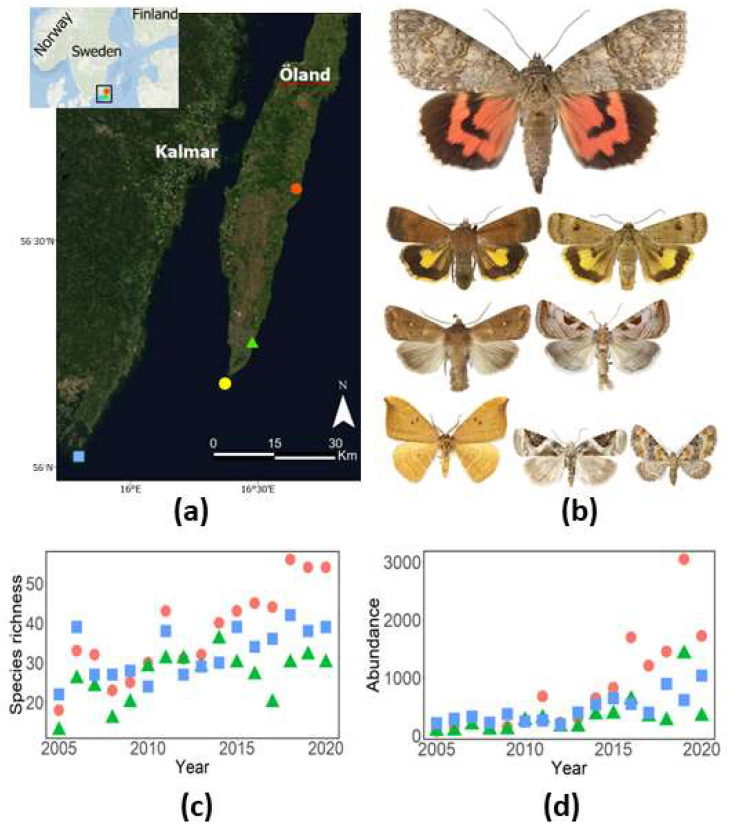
(**a**) Study area and locations of the three light-trap sites in southeastern Sweden used during the study (2005–2020), denoted as follows; Nedra Ålebäck (
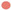
), Össby (

) and Utlängan (

). (**b**) Examples of range-expanding moths included in the study (top to bottom, left to right); *Catocala nupta*, *Noctua interjecta*, *Noctua interposita*, *Mythimna albipuncta*, *Eucarta virgo*, *Watsonalla binaria*, *Pseudeustrotia candidula* and *Eupithecia pulchellata*; photographs by Vladimir S. Kononenko. (**c**) Species richness of range-expanding moths at the three light-trap sites during the period of 2005–2020. (**d**) Abundance (expressed as the total number of individuals) of range-expanding moths at the three light-trap sites during the period of 2005–2020. The WMO meteorological station Ölands Södra udde is denoted by a yellow dot (
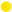
).

**Figure 2 insects-14-00055-f002:**
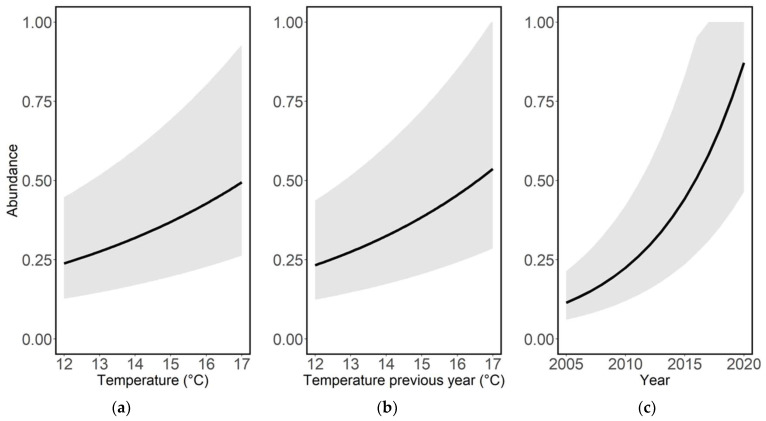
Abundance (log_10_) per species, year and site in relation to (**a**) average ambient air temperature during the recording year (t, °C), (**b**) average ambient air temperature during the previous year (t − 1, °C) and (**c**) recording year (t). Figures are based on model predictions with GLMMs and Poisson distribution (see Table 2).

**Figure 3 insects-14-00055-f003:**
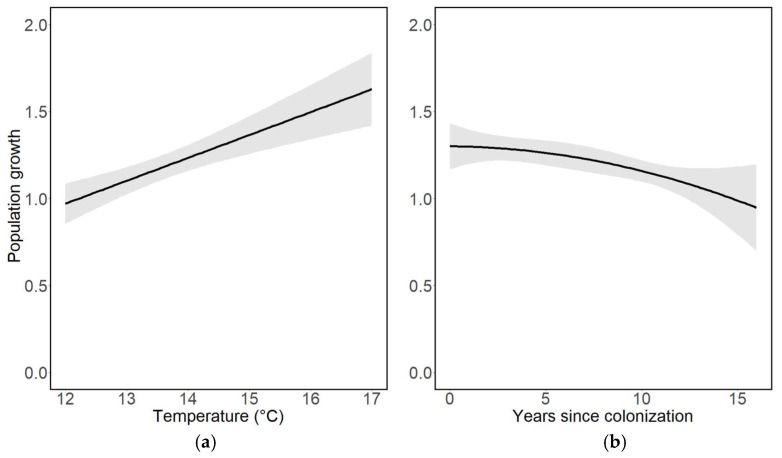
Population growth in relation to (**a**) average ambient air temperature during the recording year (t, °C) and (**b**) the number of years since colonisation of a trap site. Population growth values above 1.0 indicate positive population growth, whereas values < 1.0 indicate negative population growth (declines).

**Table 1 insects-14-00055-t001:** Results from a GLMM of species richness relative to recording year (t), average ambient air temperature during the recording year (t, °C) and average ambient air temperature during the previous year (t − 1, °C).

Predictor	Estimate	SE	*p*-Value
(Intercept)	2.175	0.682	0.01
Recording year (t)	0.035	0.006	<0.001
Temperature during recording year (t, °C)	0.048	0.03	0.112
Temperature during previous year (t − 1, °C)	0.024	0.029	0.406

**Table 2 insects-14-00055-t002:** Results from a GLMM of abundance (per species, year and site) relative to recording year (t), average ambient air temperature during the recording year (t, °C) and average ambient air temperature during the previous year (t − 1, °C).

Predictor	Estimate	SE	*p*-Value
(Intercept)	−6.517	0.357	<0.001
Recording year (t)	0.136	0.002	<0.001
Temperature during recording year (t, °C)	0.146	0.007	<0.001
Temperature during previous year (t − 1, °C)	0.167	0.006	<0.001

**Table 3 insects-14-00055-t003:** Results from a GLMM of population growth relative to the number of years since the colonisation of a trap site (linear and curvilinear, i.e., squared effects), average ambient air temperature during the recording year (t, °C) and average ambient air temperature during the previous year (t − 1, °C).

Predictor	Estimate	SE	*p*-Value
(Intercept)	−1.054	0.711	0.138
Years since colonisation	−2.574	0.799	0.003
Years since colonisation (squared) Temperature during recording year (t, °C)	−0.664 0.132	0.836 0.029	0.427 <0.001
Temperature during previous year (t − 1, °C)	0.030	0.029	0.308

## Data Availability

All data are included in the manuscript and Appendix A.

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
