# Peer review of "Associations of 16-Year Population Dynamics in Range-Expanding Moths with Temperature and Years since Establishment"

_insects, 2023, doi:10.3390/insects14010055_

Round 1

Reviewer 1 Report

In this paper, the authors analyzed light trapping data of range-extending moth species collected over 16 years in southern Sweden and found a significant increase in species richness and numbers in these regions. They concluded that the observed increase in species richness and moth numbers was due to climate warming. The results of this study are interesting. This contrasts with recent reports of extreme and widespread decreases in insect numbers and biomass. 

Overall, the manuscript is well-written, with well-illustrated results and a concise discussion. I enjoyed reading this manuscript.

Author Response

In this paper, the authors analyzed light trapping data of range-extending moth species collected over 16 years in southern Sweden and found a significant increase in species richness and numbers in these regions. They concluded that the observed increase in species richness and moth numbers was due to climate warming. The results of this study are interesting. This contrasts with recent reports of extreme and widespread decreases in insect numbers and biomass.
Overall, the manuscript is well-written, with well-illustrated results and a concise discussion. I enjoyed reading this manuscript.

Response: We thank reviewer 1 for these kind words and the appreciation of our work.

Reviewer 2 Report

Dear Authors

Thank you for your submission. The manuscript is interesting but need major changes before the further consideration. I have attached the file with comments for authors concern. 

Author Response

Point 1: Abstract is not well written. Rewrite the results section of abstract and add qualitative data too.

Response 1: We have slightly modified the abstract, and it now reads, “To inform a better understanding of population dynamics in range-expanding taxa, we studied species richness, abundance, and population growth in 25.138 individuals representing 107 range-expanding moth species at three light-trap sites in southeast Sweden over 16 years (from 2005 to 2020), in relation to temperature and years since colonisation. “

Point 2: The findings are also of applied relevance for agriculture and forestry in that they can help forecast impacts from future invasive pest species. How?

Response 2: We rephrased this sentence to clarify: ´These findings also have applied relevance for agriculture and forestry when forecasting impacts from invasive species potentially becoming serious pests within their new habitats.´

Point 3: Introduction: More attention is needed. Add new text in the respect of objectives.

Response 3: We believe this point made by the reviewer refers to the inclusion of ´adding the latest citations´ in the introduction (see point 4 below).

Point 4: you may add the latest citations too.

-https://dx.doi.org/10.17582/journal.pjz/20200112020107, and

-https://www.smujo.id/biodiv/article/view/7936

Response 4: We thank the reviewer for these kind suggestions of references of importance for our work. We have now included the first of the suggested references, ´Seasonality and climate factors….´.

Point 5: Add a table with physical characteristics of each site

Response 5: We do believe we have described the characteristics of the light-trap sites in the running text as follows, “Utlängan (56.022731 N/ 15.797629 E) is an island, 215 ha, situated 7 km southeast of the mainland (Figure 1). Habitats on the island are dominated by woods and meadows. The trap position at Utlängan is situated 3 m above sea level, 210 m from the coastline of the Baltic Sea, and semi-natural grasslands occur between the trap and the sea. Össby (56.270783 N/ 16.490312 E) and Nedra Ålebäck (56.605853 N/ 16.686114 E) are both small villages situated on the east coast of the island Öland (Figure 1). Meadows and farmlands dominate the surroundings of both villages. The trap position in Össby is situated 6 m above sea level, 370 m from the Baltic Sea, and the trap position in Nedra Ålebäck is situated 2 m above sea level and 820 m from the Baltic Sea.” We disagree with the reviewer that presenting the information about study sites in the form of a table rather than as running text would in any way improve the efficiency of information transfer. A table would however require more space.

Point 6: Results must be improved. The current statistical analysis are not much enough to describe the trend of species if 14 years?  Graphical presentations of the sites comparison is much needed?

Similarity and dissimilarity index must be added?  Authors have added the abundance trend with respect to the temperature? Why not the richness of species?

Response 6: As a reviewer, it is always possible to argue that the manuscript must be changed in different ways - but that does not mean that the analyses and results as such are necessarily wrong or in need of revision. Some reviewer comments, criticisms, and suggestions stem from misunderstandings, and some stem from the fact that the personal preferences and research interests of the reviewer(s) differ from those of the authors. In this particular case, we do not agree with the reviewer who seems to imply (i.e., “The current statistical analysis are not much enough to describe the trend of species if 14 years”) that the length of the time series studied (which is actually 16 years not 14 years as implied by the reviewer) should be insufficient to analyse trends. The existence of trends (e.g., consistent changes over time or with temperature) can of course be evaluated using both shorter and longer time series than what we have used in this study. The conclusion as to whether there is a trend or not does not depend on study duration but should be based on whether the result of the statistical analyses indicate that the observed temporal change (i.e., trend) is statistically significant such that it cannot be satisfactorily accounted for by chance alone. In our case, we do report on statistically significant changes over time, i.e., trends. It should also be emphasized here that changes in the environment and species responses have been very dramatic during the recent decade, and it is important that this is highlighted as soon as possible. As we see it, one of our paper’s strengths is that we report results from data that cover recent times (2020) instead of collecting more data and extending the time series. Further, a couple of recent studies indicate a time-span of standardised datasets of more than 10 years, able to explore population growth patterns in relation to climatic variation (Macgregor et al. 2019, Halsch et al. 2021). Further, the inclusion of three light-trap sites in this study increases the possibility of dealing with population-level stochasticity (Wagner et al. 2021).

   The suggestion of graphical representation makes partial sense, but we believe that the suggestion to compare and visualize the trap sites would be more relevant for a more specialised journal and perhaps for an in-depth study focusing on the species composition of the sites not addressed in this study. Regarding the comments about species richness trend and temperature, the analysis of species richness is significant only for the variable ´recording year´ but not for ´temperature year t´ or ´temperature year t-1´(see Table 1). To save space, as being non-significant in the analysis, we do not present a figure of this relation, but instead, we included ´species richness´ vs ´recording year´ as Figure 1c.

As for the suggestion by the reviewer that a description of the number of species per site and similarity in species composition between the sites should be included, we agree that this may be of some interest (cf. also point 7 below). We have therefore added such information in the revised version. The results section now reads, “We recorded 25.138 individuals representing 107 range-expanding moth species at the three light-trap sites over 16 years between 2005 and 2020 (Table S1). Most species were found in Nedre Ålebäck with 88 species, followed by Utlängan with 87 species and Össby with 79 species. All three sites were occupied by 58% (62 species), and 22 species only occurred at one site. Out of the 107 species, 32 were recorded for the first time in the”…

Point 7: Discussion: It will be re-structured after the results modifications.

Response 7: We have only made minor changes to the discussion as the focus of this study is the recent increase in range-expanding species during 16 years in southeast Sweden. A comparision of trap-sites and species composition is not addressed in this study.

References Response point 6, reviewer 2:

-Halsch, C.A., Shapiro, AM., Fordyce, J.A., Nice, C.C., Thorne, J.H. Waetjen, D.P. and Forister, M.L. 2021. Insects and recent climate change. PNAS 118 (2) e2002543117.

-Wagner, D.L., Wagner, Grames, E.M, Forister, M.L., Berenbaum, M.R. and Stopak, D. 2021. Insect decline in the Anthropocene: Death by a thousand cuts. PNAS 118 (2) e2023989118.

-Macgregor C.J., Williams, J.H., Bell, J.R. and Thomas, C.D. 2019. Moth biomass increases and decreases over 50 years in Britain. Nature Communication 3: 1645-1649.

Reviewer 3 Report

The manuscript presents the results of long-term observation on the number of alien or range-expanding moths caught by light traps in SE Sweden. The data are analyzed in relation to ambient air temperature and time since the establishment in the new region. The results of the study and the conclusions made by the authors are new and quite important in the contexts of climate change and of the factors influencing biological invasions. The text is clearly written, tables and figures are easily understandable. Although a number of similar studies have been conducted in different regions of the world, the present work is also interesting, in particular, because the authors (in contrast to most of other studies) showed the positive effect of potential climate change on insect the natural biodiversity, on moth species richness and abundance under natural conditions. Thus, the manuscript certainly deserves publication, although I still have some comments (see below). Unfortunately, the lines are not numbered and therefore I have to indicate page number and line number from the top (or the bottom) of the page.

Page 3, the last line of the legend to Fig. 1: I would suggest showing the sign “yellow dot” in the legend, as for light-traps.

Page 3, lines 9, 10, and 14 from the bottom: please, usual abbreviations for latitude and longitude (e.g. “56.022731 N, 15.797629 E” instead of “latitude 56.022731/ longitude 15.797629”. This is standard in scientific literature and, in addition, this allows distinguishing north from south latitude and east from west longitude.

Page 4, line 5 from the top: one bracket before “Figure S1” is missing.

Page 6, Fig. 2: please, increase font size for numbers and text along the axes.

Author Response

Response to Reviewer 3 Comments

The manuscript presents the results of long-term observation on the number of alien or range-expanding moths caught by light traps in SE Sweden. The data are analyzed in relation to ambient air temperature and time since the establishment in the new region. The results of the study and the conclusions made by the authors are new and quite important in the contexts of climate change and of the factors influencing biological invasions. The text is clearly written, tables and figures are easily understandable. Although a number of similar studies have been conducted in different regions of the world, the present work is also interesting, in particular, because the authors (in contrast to most of other studies) showed the positive effect of potential climate change on insect the natural biodiversity, on moth species richness and abundance under natural conditions. Thus, the manuscript certainly deserves publication, although I still have some comments (see below). Unfortunately, the lines are not numbered and therefore I have to indicate page number and line number from the top (or the bottom) of the page.

Response: We are grateful for these kind words from reviewer 3, and have followed the suggestions made (se specific points below).

Point 1: Page 3, the last line of the legend to Fig. 1: I would suggest showing the sign “yellow dot” in the legend, as for light-traps.

Response 1: We thank reviewer 3 for suggesting this improvement of the figure legend. We have changed accordingly.

Point 2: Page 3, lines 9, 10, and 14 from the bottom: please, usual abbreviations for latitude and longitude (e.g. “56.022731 N, 15.797629 E” instead of “latitude 56.022731/ longitude 15.797629”. This is standard in scientific literature and, in addition, this allows distinguishing north from south latitude and east from west longitude.

Response 2: We have followed the suggestion by reviewer 3 and changed the way of indicating latitude and longitude for the trap sites.

Point 3: Page 4, line 5 from the top: one bracket before “Figure S1” is missing.

Response 3: Thanks for detecting this oversight. We have inserted the missing bracket.

Point 4: Page 6, Fig. 2: please, increase font size for numbers and text along the axes.

Response 4: We have revised the font size for numbers and text along the axis in Figure 2. This indeed increases the readability of Figure 2.

Round 2

Reviewer 2 Report

Dear Authors

No more comments and the below reply by authors does not make sense in science. Arguments is authors right but not this way. 

As a reviewer, it is always possible to argue that the manuscript must be changed in different ways - but that does not mean that the analyses and results as such are necessarily wrong or in need of revision. Some reviewer comments, criticisms, and suggestions stem from misunderstandings, and some stem from the fact that the personal preferences and research interests of the reviewer(s) differ from those of the authors. In this particular case, we do not agree with the reviewer who seems to imply (i.e., “The current statistical analysis are not much enough to describe the trend of species if 14 years”) that the length of the time series studied (which is actually 16 years not 14 years as implied by the reviewer) should be insufficient to analyse trends. The existence of trends (e.g., consistent changes over time or with temperature) can of course be evaluated using both shorter and longer time series than what we have used in this study. The conclusion as to whether there is a trend or not does not depend on study duration but should be based on whether the result of the statistical analyses indicate that the observed temporal change (i.e., trend) is statistically significant such that it cannot be satisfactorily accounted for by chance alone. In our case, we do report on statistically significant changes over time, i.e., trends. It should also be emphasized here that changes in the environment and species responses have been very dramatic during the recent decade, and it is important that this is highlighted as soon as possible. As we see it, one of our paper’s strengths is that we report results from data that cover recent times (2020) instead of collecting more data and extending the time series. Further, a couple of recent studies indicate a time-span of standardised datasets of more than 10 years, able to explore population growth patterns in relation to climatic variation (Macgregor et al. 2019, Halsch et al. 2021). Further, the inclusion of three light-trap sites in this study increases the possibility of dealing with population-level stochasticity (Wagner et al. 2021).

The suggestion of graphical representation makes partial sense, but we believe that the suggestion to compare and visualize the trap sites would be more relevant for a more specialised journal and perhaps for an in-depth study focusing on the species composition of the sites not addressed in this study. Regarding the comments about species richness trend and temperature, the analysis of species richness is significant only for the variable ´recording year´ but not for ´temperature year t´ or ´temperature year t-1´(see Table 1). To save space, as being non-significant in the analysis, we do not present a figure of this relation, but instead, we included ´species richness´ vs ´recording year´ as Figure 1c.